# Optimization of GaN Bent Waveguides in the Visible Spectrum for Reduced Insertion Loss

**DOI:** 10.3390/nano15030151

**Published:** 2025-01-21

**Authors:** Wendi Li, Huiping Yin, Qian Fang, Feifei Qin, Zheng Shi, Yongjin Wang, Xin Li

**Affiliations:** GaN Optoelectronic Integration International Cooperation Joint Laboratory of Jiangsu Province, Nanjing University of Posts and Telecommunications, Nanjing 210003, China; 1224014116@njupt.edu.cn (W.L.); 1223014316@njupt.edu.cn (H.Y.); 1222014633@njupt.edu.cn (Q.F.); qinfeifei@njupt.edu.cn (F.Q.); shizheng@njupt.edu.cn (Z.S.); wangyj@njupt.edu.cn (Y.W.)

**Keywords:** GaN waveguide, bent waveguide, insertion loss, back-side thinning, visible light communication, photonic integrated chips

## Abstract

The development of GaN-based photonic integrated chips has attracted significant attention for visible light communication systems due to their direct bandgap and excellent optical properties across the visible spectrum. However, achieving compact and efficient light routing through bent waveguides remains challenging due to high insertion losses. This paper presents a comprehensive investigation of GaN bent waveguides optimization for visible light photonic integrated chips. Through systematic simulation analysis, we examined the effects of bending angle, process optimization approaches, and geometric parameters on insertion loss characteristics. The back-side thinning process demonstrates superior performance compared to front-side etching, reducing the insertion loss of 90° bends from 1.80 dB to 0.71 dB. Further optimization using silver reflection layers achieves an insertion loss of 0.57 dB. The optimized structure shows excellent performance in the blue-green spectral range (420–500 nm) with insertion losses below 0.9 dB, providing practical solutions for compact GaN photonic integrated chips in visible light communications.

## 1. Introduction

The exponential growth in data traffic has driven the exploration of new communication technology beyond radio frequencies [1,2]. Visible Light Communication (VLC), leveraging the vast frequency spectrum between 400–800 THz, has emerged as a promising technology for future communication systems [3,4]. VLC not only solves spectrum limitations but also provides advantages in electromagnetic interference-sensitive environments such as medical facilities and aviation [5,6]. With the development of advanced modulation techniques and device optimization, VLC systems have demonstrated multi-Gbps transmission rates, showing great potential to support the high-speed transmission requirements of future 6G communications [7,8].

The III-nitride material system, particularly GaN and others, offers remarkable advantages for visible light photonic integrated chips due to its direct bandgap and excellent optical properties [9,10]. Through bandgap engineering of GaN (3.4 eV) and AlN (6.2 eV), optical devices can be designed to cover the entire visible spectrum [11]. The integration capability of GaN-based materials provides solutions for realizing compact and efficient communication systems through monolithic integration of emitters, modulators, waveguides, detectors, and other functional components [12,13,14].

For practical photonic integrated chips, bent waveguides play a crucial role in enabling flexible light path control and achieving compact chip design [15,16]. While straight waveguides support transmission paths, bent waveguides are essential for complex connection to realize higher integration density and flexible light signal routing [17,18]. The integration of multiple optical components on a single GaN chip presents compact, energy-efficient VLC transceivers [19]. However, bent waveguides face unique challenges such as radiation losses and mode mismatch at curved parts [20,21]. It is needed to provide careful design and innovative fabrication approaches. The development of high-performance optical interconnects in GaN-based photonic integrated chips requires optimization of bent waveguides to achieve efficient light connection with low losses.

In the infrared wavelength range, significant progress has been achieved in research of low-loss bent waveguides. For a silicon-based platform, Vlasov and McNab demonstrated low-loss waveguide bends with a bending loss of 0.086 ± 0.005 dB/turn at a bending radius of 1 μm through optimization of the fabrication process [22]. Recently, Zhang et al. achieved even more compact design using air trenches and an embedded germanium arc, reducing the bending radius to below 500 nm while maintaining an insertion loss as low as 0.12 dB and broadband operation over 500 nm of bandwidth [16].

For higher integration density platforms, notable progress has been demonstrated using multi-layer integration. Gao et al. demonstrated compact silicon nitride and silicon waveguide bends using a CMOS-compatible SiN-on-SOI platform over the C band [23]. By employing linearly changing curvature, they achieved significant bending loss reduction to 0.037 dB/90° with a 30 μm radius for a PECVD SiN waveguide. For medium index contrast waveguides, Środa et al. proposed a novel two-step thickness structuring method, achieving inter-mode crosstalk below −40 dB with bending a radius of 100 μm [24].

Various optimization approaches have been demonstrated, including air trenches [16], subwavelength gratings [25], inverse design [26], and gradual changes in waveguide width and curvature [27]. The latest research indicates a sub-micron bending radius is achievable through proper design optimization. However, for GaN-based bent waveguides operating in visible wavelengths, systematic research is still limited, indicating significant value for future investigation.

While significant progress has been achieved in GaN-based straight waveguides [28,29], research on GaN bent waveguides remains limited, particularly in the visible spectrum. Initial demonstrations have shown promising potential for integrated photonic applications. Cai et al. achieved 70 Mbps data transmission using integrated GaN bent waveguides on silicon [30], and Li et al. demonstrated monolithically integrated LEDs, photodetectors, and bent waveguides on both silicon and sapphire substrates [31,32]. However, current studies mainly focus on specific wavelengths and limited geometrical parameters. Sekiya et al. reported waveguide losses of 2.6 dB/mm at 406 nm for straight waveguides [33], but comprehensive characterization of bent waveguide performance across different bending angles, radii, and wavelengths remains unexplored.

Despite these advances, several critical gaps remain in the current studies of GaN bent waveguides for visible light communication systems. While previous studies have made important progress in specific aspects, such as Sekiya et al. [33] reporting waveguide losses of 2.6 dB/mm at 406 nm for straight waveguides and Li et al. [31,32] demonstrating monolithically integrated LEDs with bent waveguides for VLC applications, systematic analysis covering the full visible spectrum (380–800 nm) is still lacking. This limitation has hindered understanding of spectral dependent performance across different wavelengths, which is crucial for VLC system design. The relationship between geometric parameters and insertion loss requires deeper investigation, particularly how varying bending angles (5–90°) impact waveguide performance in integrated VLC circuits. The current studies typically focus on fixed angles like 90° without the comprehensive angular analysis needed for flexible VLC chip layouts. Although various fabrication processes have been reported, quantitative comparisons between different techniques like front-side etching and back-side thinning remain insufficient for establishing optimal VLC chip processing guidelines. While both approaches have been demonstrated separately [34,35], their relative effectiveness in reducing insertion loss has not been systematically compared. Additionally, while material studies show promise for performance enhancement, the applications of metal reflection layers for loss reduction in GaN bent waveguides remains unexplored, especially regarding the evaluation of different metals’ performance in the visible spectrum where VLC systems operate. Previous studies in silicon photonics [16,22] have shown that metal claddings can significantly reduce bending losses in the infrared region, but similar systematic studies for GaN waveguides in the visible spectrum, which is essential for VLC applications, have not been carried out.

The unique material properties and fabrication challenges of GaN necessitate further systematic investigation. The epitaxial growth on foreign substrates introduces high dislocation densities and crystal defects, making precise control of waveguide geometry and surface quality critical for optimal performance [36]. Existing research lacks comprehensive modeling of loss mechanisms specific to GaN bent waveguides in the visible spectrum. There is an urgent need to establish complete loss models across different wavelengths and develop innovative fabrication approaches for reducing insertion losses. Additionally, systematic process optimization studies are required to address the stronger optical confinement at visible wavelengths and minimize radiation losses at bent sections. These fundamental understandings will enable the development of high-performance GaN photonic integrated chips operating in the visible spectrum.

Our group has made significant progress in GaN-based photonic integration research. We have successfully demonstrated bent waveguides in GaN-on-silicon platform with monolithically integrated LEDs, achieving 100 Mbps visible light communication [35]. Through systematic investigation, we found that geometrical parameters optimization of bent waveguides (width reduction from 8 μm to 4 μm, curvature diameter from 500 μm to 250 μm) significantly impacts optical transmission loss characteristics. Moreover, we have developed innovative process optimization approaches including back-side thinning and metal reflection layers for suspended waveguide structures [36,37]. These process innovations have effectively enhanced the waveguide transmission efficiency and established the foundation for complex photonic device integration.

However, several critical challenges remain to be addressed. The comprehensive quantitative analysis of transmission characteristics under various bending angles (5–90°) is still lacking. The correlation between loss mechanisms and process parameters needs to be established through rigorous theoretical modeling. Additionally, novel approaches such as metal reflection layers require in-depth investigation of their effects on bent waveguide performance. Therefore, systematic studies combining experimental characterization and theoretical simulation are essential to achieve quantitative understanding and optimization of GaN bent waveguides [38].

Recent advances in nanophotonic devices also presented the crucial role of material engineering and structural optimization for improving photodetection capabilities. For instance, chemically doped graphene-based field-effect transistors have shown remarkable improvements in photoresponsivity through controlled carrier modulation [39]. Similarly, novel two-dimensional materials like black arsenic have exhibited promising polarization-sensitive photodetection properties with significant anisotropic responses [40]. These developments in nanoscale photodetectors highlight the importance of optimizing both material properties and device geometries for improved performance. In the context of GaN-based photonic integrated circuits, such optimization principles are equally crucial, particularly for bent waveguides that serve as fundamental building blocks for complex optical routing and signal processing. The optimization of bent waveguides directly impacts the overall performance of integrated photodetectors and other optoelectronic components in the visible spectrum.

Based on the current challenges and our previous findings in GaN-based bent waveguides, this work presents a comprehensive study on optimizing insertion loss characteristics in the visible spectrum through both process innovations and geometric parameter optimization. We first establish a systematic optical simulation model to analyze the loss mechanisms in GaN bent waveguides. Two distinct process optimization approaches—front-side etching and back-side thinning—are proposed and comparatively investigated to address the slab layer-induced losses. The effects of critical geometric parameters, including waveguide width, sidewall angle, and bending radius, are thoroughly examined through finite element simulations. Furthermore, we explore the potential of metal reflection layers for additional loss reduction, with various metals evaluated for their effectiveness. Notably, our optimized structure achieves an insertion loss as low as 0.57 dB for 90-degree bends in the blue-green spectral range, representing a significant improvement over the conventional design. The comprehensive characterization across the visible spectrum (380–800 nm) provides valuable insights for designing high-performance GaN photonic integrated circuits. These findings establish practical guidelines for realizing compact, low-loss bent waveguides essential for next-generation visible light communication systems.

## 2. Optical Simulation Model

We established an optical simulation model to analyze and optimize the insertion loss of GaN bent waveguides in the visible spectrum. Bent waveguides play a crucial role in photonic integrated chips by enabling flexible routing of optical signals and reducing the footprint of photonic chips. The study is based on GaN epitaxial layers grown on silicon substrate, with a GaN-based epitaxial layer thickness of 4.375 μm (as illustrated in the cross-section view of the GaN-based epitaxial layer on a silicon substrate in Figure 1a).

We employed the Beam Propagation Method (BPM), which is based on solving the paraxial Helmholtz equation, to analyze the optical field propagation in the GaN bent waveguides. The scalar wave equation for the electric field E(x,y,z) can be written as follows:∇^2^E + k_0_^2^n^2^(x,y)E = 0(1)
where k_0_ = 2π/λ is the wavenumber in vacuum and n(x,y) is the refractive index distribution in the waveguide cross-section. For the bent waveguide analysis, we apply the conformal transformation to map the curved waveguide to an equivalent straight waveguide with modified refractive index distribution using the following equation:n′(x,y) = n(x,y)exp(x/R)(2)
where R is the bending radius. The field is expressed using the slowly varying envelope approximation:E(x,y,z) = ψ(x,y,z)exp(−jk_0_n_0_z)(3)
where n_0_ is the reference refractive index. Substituting this into the wave equation and applying the paraxial approximation leads to the BPM equation:∂ψ/∂z = [j/(2k_0_n_0_)]∇^2^ₜψ + jk_0_(n^2^(x,y) − n_0_^2^)ψ/2n_0_(4)
where ∇^2^ₜ is the transverse Laplacian operator. This equation is solved numerically using the split-step Fourier method:ψ(x,y,z + Δz) ≈ exp(jk_0_Δz(n^2^ − n_0_^2^)/2n_0_)exp(jΔz∇^2^ₜ/2k_0_n_0_)ψ(x,y,z)(5)

The simulation implements the BPM algorithm through a series of computational steps. The process begins with initializing a Gaussian beam profile that matches the fundamental mode of the input waveguide. As the optical field propagates along the z-direction, the algorithm applies alternating steps in real and Fourier space to account for both refractive index variations and diffraction effects. At each propagation step, the phase changes due to the refractive index profile are calculated in real space, while the diffraction effects are more efficiently computed in the frequency domain using Fast Fourier Transforms. The guided mode power at any position z is obtained by calculating the overlap integral between the propagating field ψ(x,y,z) and the fundamental mode profile ϕ(x,y), expressed as P(z) = |∫∫ψ*(x,y,z)ϕ(x,y)dxdy|^2^, where ϕ(x,y) is the fundamental mode profile of the waveguide. The insertion loss in decibels is then computed as follows:Loss (dB) = −10log_10_(P(L)/P(0))(6)
where L is the total propagation length. The simulation parameters were set with a transverse grid spacing of λ/20 and longitudinal step size of λ/4 to ensure numerical accuracy. The computational window was chosen to be five times the waveguide width with perfectly matched layer boundary conditions.

The BPM method is particularly suitable for analyzing GaN bent waveguides in the visible spectrum since the paraxial approximation remains valid with our design’s maximum bend angle relative to the propagation direction being below 15° in the 90° bent structure. The conformal transformation technique accurately accounts for the curved geometry effects while maintaining computational efficiency. The split-step implementation allows separate handling of refractive index variation and diffraction effects, which is essential for modeling the complex refractive index distribution in our GaN-based epitaxial layers.

The validity of this simulation approach has been verified through comparison with experimental results from our previous studies on GaN bent waveguides. In our earlier work [35], we experimentally demonstrated bent waveguides in a GaN-on-silicon platform with monolithically integrated LEDs, achieving 100 Mbps visible light communication. The measured insertion loss was 1.85 dB for 90-degree bends with 10 μm radius, showing excellent agreement with this current simulation prediction of 1.80 dB under identical geometric conditions. This agreement has also been further validated by independent experimental studies from our group, where Cai et al. [30] achieved successful data transmission using integrated GaN bent waveguides with comparable performance characteristics. For straight waveguides, our model predictions align well with the experimental results reported by Sekiya et al. [33], who demonstrated propagation losses of 2.6 dB/mm at 406 nm. Additional experimental studies by Gromovyi et al. [28] and Chen et al. [29] on GaN waveguides in the visible spectrum range have shown similar loss characteristics, further supporting the reliability of our simulation approach.

The bent waveguide was fabricated on the GaN epitaxial layer using microfabrication techniques. The geometric parameters of the bent waveguide without process optimization are as follows: waveguide width of 2 μm, height of 1 μm, and bending radius of 10 μm with a 90-degree bend angle and a slab layer thickness of 3.375 μm under the bent waveguide. These parameters were selected based on our research group’s reported work. The GaN epitaxial layer on a silicon substrate is typically thick (approximately 4–5 μm), and the processing precision of lithography and III-V material ICP etching technology is limited for the waveguide of the photonics chip. Furthermore, to minimize the footprint of bent waveguides on photonic chips, the waveguide bending radius and width were specifically designed to the aforementioned dimensions. Figure 1a shows the cross-section view of the GaN-based epitaxial layer on a silicon substrate, and Figure 1b illustrates the schematic diagram of the bent waveguide without process optimization.

In the bent waveguide without process optimization, the thick slab layer leads to high order mode leakage into the slab region. When the optical field penetrates the slab layer, the refractive index contrast between the slab layer and substrate causes optical leakage through their interface, resulting in insertion loss as the leaked light is absorbed by the substrate. To solve this problem, we propose two process optimization approaches: front-side etching and back-side thinning processes. The front-side etching process directly removes the slab layer from the top surface, while the back-side thinning process eliminates light leakage by removing the silicon substrate and slab layer from the back-side. Our simulation results demonstrate that while the bent waveguide without process optimization exhibits significant optical mode distribution in the slab region, the optimization processes successfully confine the optical field within the waveguide core. We employ the Beam Propagation Method (BPM)-based finite element simulation to analyze and compare the improvement effects of these two optimization processes. Additionally, this study systematically investigates the impact of various parameters and processes on insertion loss, including sidewall inclination and metal deposition on the waveguide.

Figure 2a shows the front-side etching process flow, which consists of the following steps: (a) a photolithography process to pattern the bent waveguide structure; (b) transfer of the waveguide pattern from the photoresist to GaN-based epitaxial layer using III-V material ICP (Inductively Coupled Plasma) etching; (c) ICP etching to thin and remove the slab layer beneath the bent waveguide; and (d) removal of residual photoresist to obtain the bent waveguide. Figure 2b illustrates the back-side thinning process flow, including: (a) a photolithography process to pattern the bent waveguide structure; (b) transfer of the pattern to GaN layer using ICP etching; (c) removal of the silicon substrate beneath the bent waveguide using back-side photolithography and STS (Silicon Through-Substrate Etching); and (d) thinning and removal of the slab layer from the backside of the waveguide using III-V material ICP etching. Figure 2c presents the cross-sectional views of the bent waveguides fabricated by these two processes with the following key geometric parameters: width W, height H, slab layer thickness T (beneath the bent waveguide), and sidewall angle α. The sidewall angle α is introduced in the simulation analysis to account for potential sidewall inclination caused by the ICP etching process and its impact on waveguide insertion loss.

We used the Beamprop module of the Rsoft optical simulation software to analyze the insertion loss of GaN bent waveguides based on the Beam Propagation Method. The optimization process was conducted through systematic parameter sweeping in the Beamprop environment. For each design parameter (waveguide width, slab thickness, sidewall angle, etc.), we performed sequential optimization by varying one parameter while keeping others constant to identify the optimal value. The software’s built-in optimization algorithm uses the finite difference beam propagation method (FD-BPM) with an alternating direction implicit (ADI) scheme for efficient 3D waveguide analysis. The optimization objectives were set to minimize insertion loss while maintaining practical fabrication constraints. All simulations were performed using Rsoft CAD Version 2021.09 with standard single-processor computation on a workstation with 64 GB RAM. Before using front-side etching or back-side thinning process optimization, the geometric parameters of the bent waveguide model were set as follows. The waveguide width W was 2 μm, height H was 1 μm, and the bending radius was 10 μm with a 90-degree bend angle. The initial slab layer thickness T under the bent waveguide was 3.375 μm, and the sidewall angle α was set to 0 degrees. The wavelength of visible light in the model was set to 450 nm (in the blue range). The GaN-based epitaxial layer grown on a 200 μm silicon substrate consists of 300 nm AlN, 700 nm AlGaN, 600 nm GaN, 2600 nm n-type GaN, 60 nm InGaN/GaN multiple quantum wells, 35 nm p-type AlGaN, and 80 nm p-type GaN. This epitaxial layer structure was selected based on our research group’s reported work. The refractive indices of these layers were set according to the built-in material database of the Rsoft software and an online refractive index database. The refractive index distribution of the cross-section is shown in Figure 3.

## 3. Analysis of Waveguide Loss Characteristics

A visible Gaussian light source with a relative power of 1.0 is set as the excitation source at the input end of the GaN bent waveguide. The optical power along the waveguide is monitored relative to its transmission through the bent waveguide, with different bending angles. Figure 4 shows the relationship between bending angle and insertion loss of the GaN bent waveguide without optimization. The insertion loss increases gradually with the bending angle from 5° to 90°. At small bending angles (5–20°), the insertion loss increases slowly from 0.24 dB to 0.49 dB. As the bending angle increases to the medium range (25–60°), the loss increases from 0.70 dB to 1.27 dB. The most significant increase occurs at larger angles (65–90°), where the insertion loss reaches 1.80 dB at 90°. This angle-dependent loss analysis reveals the sensitivity of our GaN-based material system to bending effects, particularly at larger angles where mode mismatch and radiation losses become more significant. The sharp increase in insertion loss at angles above 60° indicates a critical threshold where the waveguide’s guiding mechanism becomes less effective, due to enhanced mode coupling to radiation modes at the bend. Understanding this concept is crucial for designing photonic integrated chips where 90° bends are often required for compact chip design. The relatively high insertion loss at 90° (1.80 dB) suggests the necessity for process optimization, especially considering that photonic chips may require multiple bent waveguides for signal routing. Furthermore, this baseline characterization provides quantitative targets for subsequent optimization strategies, which aim to maintain the compactness of 90° bends while reducing insertion losses.

Figure 5a illustrates that in the 90-degree bent waveguide without optimization, the optical power within the waveguide experiences a substantial decrease with increasing transmission distance, reducing to 0.661 (1.8 dB insertion loss) at the waveguide’s output end. The insertion loss in the bent waveguide originates from both the bending effect and the leakage through the 3.375 μm thick slab layer and silicon substrate. The combined effect leads to significant optical mode leakage into the surrounding medium, particularly in the bent region. To reduce insertion loss and improve the performance of integrated photonic chips, two process optimization approaches were investigated. The front-side etching that directly removes the slab layer from the top surface, and back-side thinning that eliminates the leakage by removing both the silicon substrate and slab layer from backside. As shown in Figure 5b, the front-side etching process improves the relative optical power at the output end to 0.751 (1.2 dB insertion loss) by removal of the slab layer. More notably, as shown in Figure 5c, the back-side thinning process achieves even better performance with a relative optical power of 0.849 (0.71 dB insertion loss) at the output end. The superior performance of the back-side thinning process can be attributed to its comprehensive approach in eliminating the leakage path while maintaining better waveguide structural integrity. The back-side thinning process not only significantly reduces insertion loss but also promotes more uniform optical mode distribution within the waveguide core, which is particularly beneficial for bent waveguides.

Figure 6 shows the relationship between slab layer thickness and insertion loss under two different process optimization approaches. In the bent waveguide without optimization, the insertion loss is 1.80 dB with a slab layer thickness of 3.375 μm. The back-side thinning process begins with the removal of the silicon substrate through STS etching, which immediately reduces the insertion loss to 1.01 dB, followed by ICP etching to thin the suspended GaN-based epitaxial slab layer. As the slab layer is gradually thinned, the insertion loss continues to decrease, ultimately reaching 0.71 dB when the slab layer is completely removed. In contrast, the front-side etching process shows less improvement, with the insertion loss decreasing from 1.80 dB to 1.20 dB as the slab layer thickness is reduced. When the slab layer thickness exceeds 1.875 μm, the back-side thinning process demonstrates better performance compared to the front-side etching process, with a difference in insertion loss of up to 0.85 dB. As the slab layer thickness decreases below 1.875 μm, both of the processes reduce insertion loss, but the back-side thinning process has superior performance throughout the whole range. The enhanced performance of the back-side thinning process can be attributed to its two-step approach, first eliminating the substrate absorption, then removing the slab layer, which more effectively suppresses optical mode leakage in the bent region of the waveguide.

During the fabrication of GaN bent waveguides using III-V material ICP dry etching technology, sidewall inclination is an inevitable phenomenon that affects waveguide performance. To quantify this effect in the back-side thinning optimized waveguide structure, we investigated the relationship between sidewall angle and insertion loss. The sidewall angle α was varied from 0° (perfectly vertical) to 45° in 5° increments, where this range represents typical sidewall angles achievable with the current ICP etching processes. As shown in Figure 7, for small sidewall angles (0–15°), the insertion loss shows a moderate increase from 0.71 dB to 0.98 dB, indicating that slight deviations from vertical sidewalls can be tolerated. In the intermediate range (20–30°), the rate of increase slows, with the insertion loss rising from 1.08 dB to 1.13 dB. An obvious increase is observed for larger angles (35–45°), where the insertion loss reaches 1.39 dB. This enhanced loss at larger sidewall angles can be attributed to two primary mechanisms: the increased scattering at the non-vertical interfaces and modification of the optical mode distribution in the bent region. The non-vertical sidewalls effectively increase the interaction length between the guided mode and the waveguide boundaries, particularly critical in the bent section where the mode is already displaced towards the outer radius. Additionally, the inclined sidewalls alter the effective cross-sectional geometry of the waveguide, leading to enhanced mode mismatch and subsequent radiation losses in the bent region. These results present the importance of optimizing ICP etching parameters to achieve near-vertical sidewalls for minimizing insertion losses in GaN bent waveguides.

To further reduce the insertion loss of the bent waveguide after back-side thinning optimization, we investigated the effect of depositing different metal films on the backside of the waveguide. In bent waveguides, the optical mode tends to shift towards the outer radius of the bend, potentially leading to radiation losses at the waveguide boundaries. A metal film on the waveguide backside can serve as a reflective boundary to confine the optical mode and suppress radiation losses. The metals selected for this study (Ag, Al, Au, Cr, Ni, and Ti) are commonly used in semiconductor fabrication processes. Figure 8 shows the insertion loss of GaN bent waveguides with various 100-nm-thick metal films deposited on the waveguide backside. The bent waveguide optimized by back-side thinning has an insertion loss of 0.71 dB. With Ag and Al films, the waveguides demonstrate better performance, reducing the insertion loss to 0.57 dB and 0.59 dB, respectively. This improvement can be attributed to their high reflectivity in the visible range, which effectively confines the optical mode within the waveguide core and reduces radiation losses at the bend. In contrast, Au, Cr, Ni, and Ti films lead to increased insertion losses of 1.27 dB, 1.12 dB, 1.21 dB, and 1.12 dB, respectively. The degraded performance with these metals likely results from their stronger optical absorption in the visible range, which reduce their reflective effect. Ag shows the optimal performance with a 19.7% reduction in insertion loss, making it the most promising candidate for metal-backed GaN bent waveguides in practical photonic integrated chips.

The distinct performance differences among these metals could be attributed to their different optical properties in the visible spectrum [41,42]. Ag and Al exhibit superior performance due to their high reflectivity and low optical absorption in visible range, with Ag reflectivity above 95% and Al above 90% at 450 nm [41]. Their high plasma frequencies (Ag: ~9.2 eV, Al: ~12.7 eV) and low optical losses make them fine reflectors for visible light. This high reflectivity effectively confines the optical mode within the waveguide structure, reducing radiation losses at the bend part.

In contrast, Au has higher insertion loss despite being a good conductor. This phenomenon could be explained by its characteristic inter band transitions in the visible spectrum, particularly below 500 nm, where d-band to s-band transitions lead to strong optical absorption [42]. Au’s plasma frequency (~8.9 eV) and complex dielectric function result in lower reflectivity and higher absorption in the blue and green range compared to Ag and Al. Similarly, the transition metals (Cr, Ni, and Ti) also have higher losses due to their partially filled d-bands, which enable numerous electronic transitions and consequently higher optical absorption in the visible range [41]. The selection of Ag as the optimal reflective material is supported by its fundamental optical properties in the visible spectrum. Its high plasma frequency (~9.2 eV) and low optical losses make it an excellent reflector at visible wavelengths, with reflectivity exceeding 95% at 450 nm [41,42], enabling effective optical mode confinement in the bent waveguide structure.

The width of the waveguides significantly influences both optical mode and fabrication quality. We studied the relationship between waveguide width and insertion loss for the back-side thinning optimized bent waveguide. Figure 9 shows the insertion loss with waveguide width ranging from 0.2 μm to 2.0 μm. For widths 0.2–0.4 μm, despite their potential advantage in realizing fewer modes or even single-mode transmission, the waveguides have high insertion losses of 1.42 dB and 1.28 dB, respectively. The higher loss in narrow waveguides stems from enhanced evanescent field interaction with boundary imperfections and increased sensitivity to fabrication defects, particularly in the etched sidewalls where nanometer-scale roughness becomes more critical. In the width range 0.6–1.0 μm, the insertion loss is around 1.27–1.21 dB, presenting a balance between mode confinement and fabrication tolerance. A significant improvement in performance is observed as the width increases beyond 1.2 μm, with the insertion loss decreasing from 1.15 dB to 0.71 dB at 2.0 μm. The loss reduction results from enhanced mode confinement and better mode distribution in the bent region. While the 2.0 μm width demonstrates a good insertion loss, the wider waveguides support multiple modes. The choice of waveguide width must balance many factors: insertion loss, mode control, fabrication tolerance, and the space constraints in photonic integrated chips.

To evaluate the spectral performance of the optimized GaN bent waveguide across the entire visible spectrum, we extended the above analysis to cover the wavelength range from 0.34 μm to 0.80 μm, providing comprehensive characterization with adequate margins beyond the visible range. Figure 10 shows the wavelength dependence of insertion loss for the back-side thinning optimized bent waveguide, which exhibits distinct behaviors in different spectral regions. In the ultraviolet region (0.34–0.38 μm), extremely high insertion losses exceeding 23 dB are observed. The great loss originates from GaN’s material properties, specifically its band-edge absorption as the wavelength approaches GaN’s bandgap energy (3.4 eV). The strong absorption in this region makes waveguiding challenging, which is consistent with theoretical predictions based on GaN’s electronic band structure. A dramatic transition occurs between 0.38–0.42 μm, where the insertion loss sharply decreases from 7.26 dB at 0.40 μm to 0.59 dB at 0.42 μm. The rapid reduction is consistent with GaN’s fundamental absorption edge, marking the transition from high loss to low loss. The waveguide achieves optimal performance in the blue-green range (0.42–0.50 μm) with insertion losses below 0.9 dB. The superior performance may be attributed to the optimal refractive index contrast between GaN and air in this range, which provides strong mode confinement. The Ag reflection layer also exhibits peak reflectivity in this spectral region, maximizing its effectiveness in reducing radiation losses. Beyond a 0.50 μm wavelength, the insertion loss gradually increases, reaching 2.43 dB at 0.80 μm. The degradation can be understood through the wavelength-dependent mode behavior. As the wavelength increases, the effective mode area expands, leading to reduced mode confinement and increased evanescent field penetration into the surrounding medium. The larger mode size at longer wavelengths results in enhanced bending radiation losses, as the optical field has greater distortion in the bent region.

The wavelength-dependent characteristics highlight the importance of considering spectral effects in GaN waveguide design. The optimized structure is particularly well-suited for applications in the blue-green range, making it an excellent candidate for visible light communication systems operating in this wavelength region. For applications requiring operation at wider wavelengths, additional geometric optimization or alternative design strategies may be necessary to maintain low insertion losses.

## 4. Conclusions

This work has systematically investigated the optimization of GaN bent waveguides through process innovation and geometric parameter optimization. The bending angle-dependent analysis reveals that conventional structures exhibit increasing losses from 0.24 dB at 5° to 1.80 dB at 90°. The back-side thinning process demonstrates superior performance by reducing the 90° bend loss by 60.6% to 0.71 dB, significantly outperforming front-side etching optimization. Geometric parameter analysis shows critical dependence on sidewall angle and waveguide width. Maintaining sidewall angles below 15° is crucial, as the insertion loss increases from 0.71 dB to 0.98 dB at 15°, and further deteriorates to 1.39 dB at 45°. Waveguides wider than 1.2 μm exhibit substantially improved performance, though this must be balanced against potential multimode operation. The integration of 100-nm thick silver reflection layers provides additional improvement, reducing insertion loss by 19.7% to 0.57 dB. Spectral characterization demonstrates optimal performance in the blue-green region (420–500 nm) with losses consistently below 0.9 dB. These findings establish practical design guidelines for GaN bent waveguides in visible light photonic integrated chips and suggest future directions in experimental validation and advanced optimization techniques.

## Figures and Tables

**Figure 1 nanomaterials-15-00151-f001:**
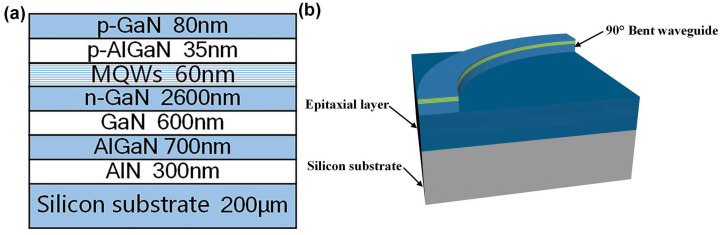
(**a**) Cross-section view of the GaN-based epitaxial layer on a silicon substrate; and (**b**) schematic diagram of the bent waveguide without process optimization.

**Figure 2 nanomaterials-15-00151-f002:**
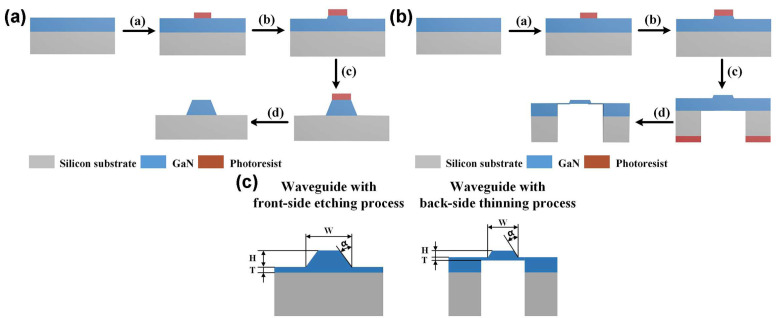
(**a**) The front-side etching process flow of the bent waveguide; (**b**) the back-side thinning process flow of the bent waveguide; and (**c**) the cross-sections of the waveguides prepared by the two processes.

**Figure 3 nanomaterials-15-00151-f003:**
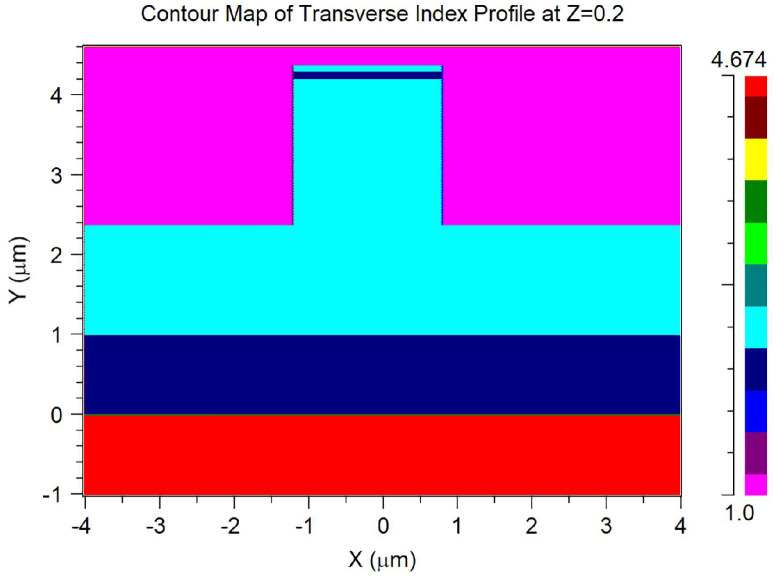
Geometrical structure and refractive index distribution of the cross-section of the bent waveguide without optimization.

**Figure 4 nanomaterials-15-00151-f004:**
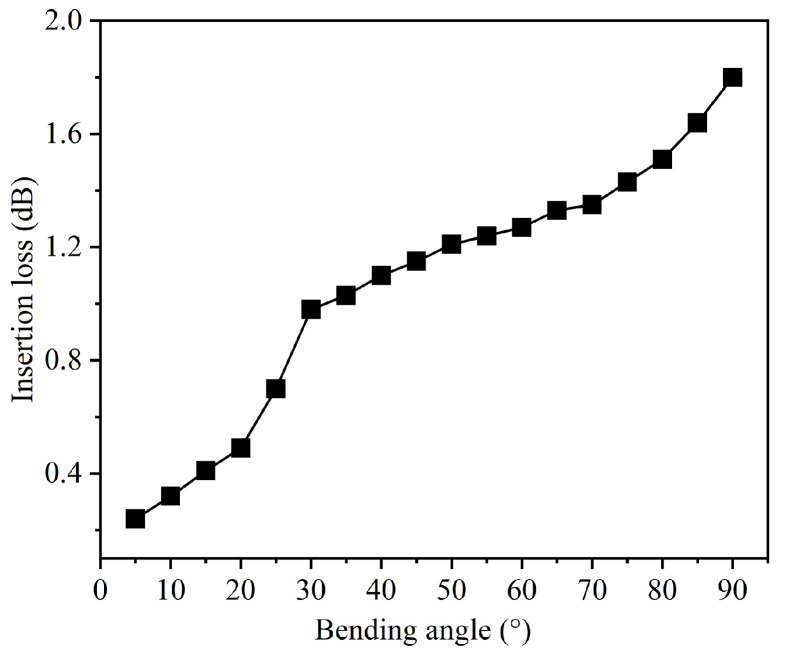
Insertion loss of the GaN bent waveguide without optimization and with different bending angles.

**Figure 5 nanomaterials-15-00151-f005:**
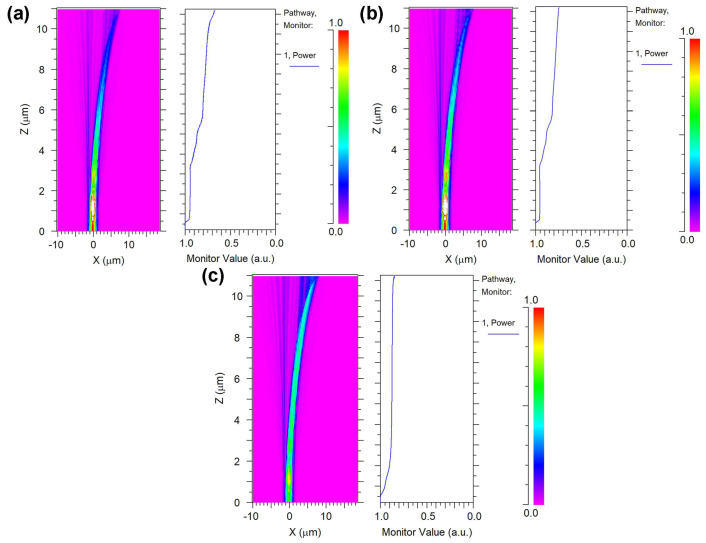
Relative light output distribution of the GaN bent waveguide: (**a**) 90-degree bent waveguide without optimization; (**b**) front-side etching process removes the slab layer; and (**c**) back-side thinning process removes the silicon substrate and slab layer.

**Figure 6 nanomaterials-15-00151-f006:**
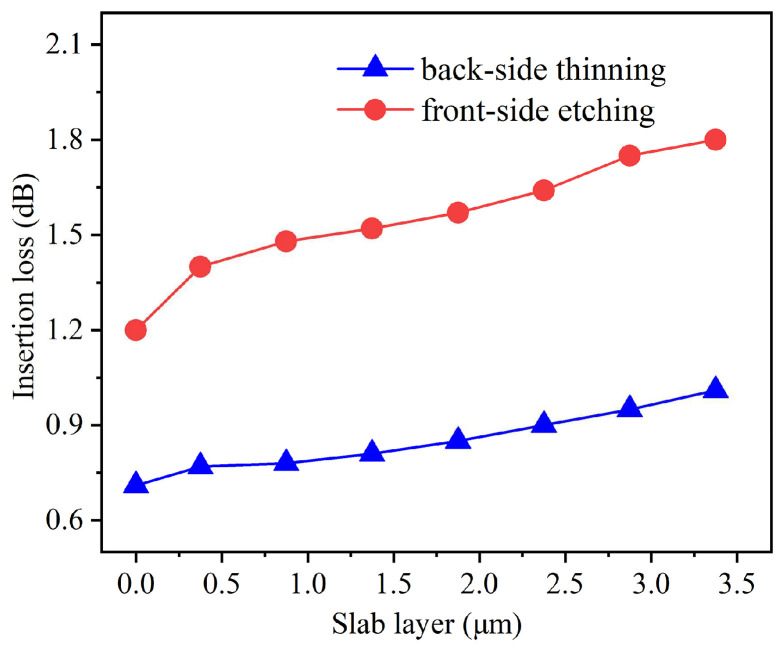
Insertion loss of the GaN bent waveguide with different slab layer thickness under two optimization processes.

**Figure 7 nanomaterials-15-00151-f007:**
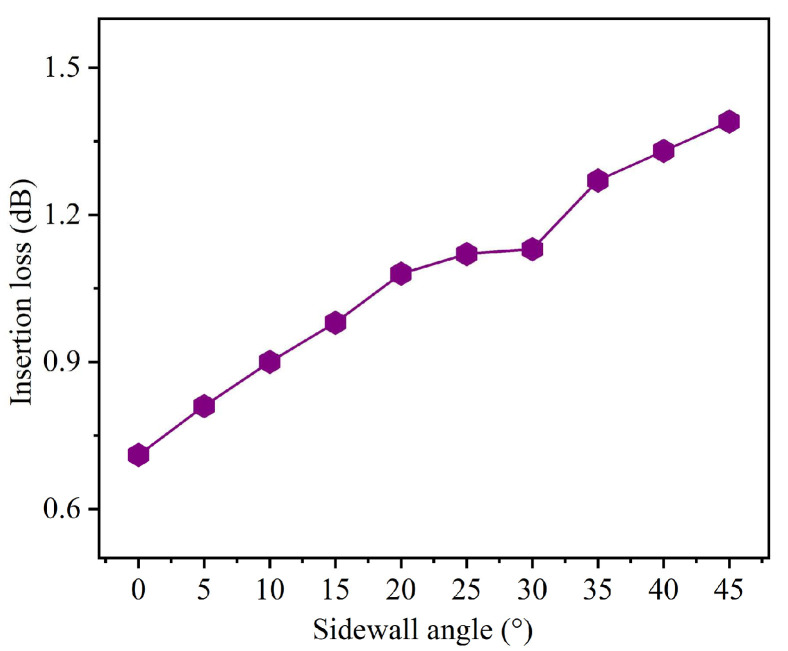
Insertion loss of the GaN bent waveguide with different sidewall angles under back-side thinning optimization processes.

**Figure 8 nanomaterials-15-00151-f008:**
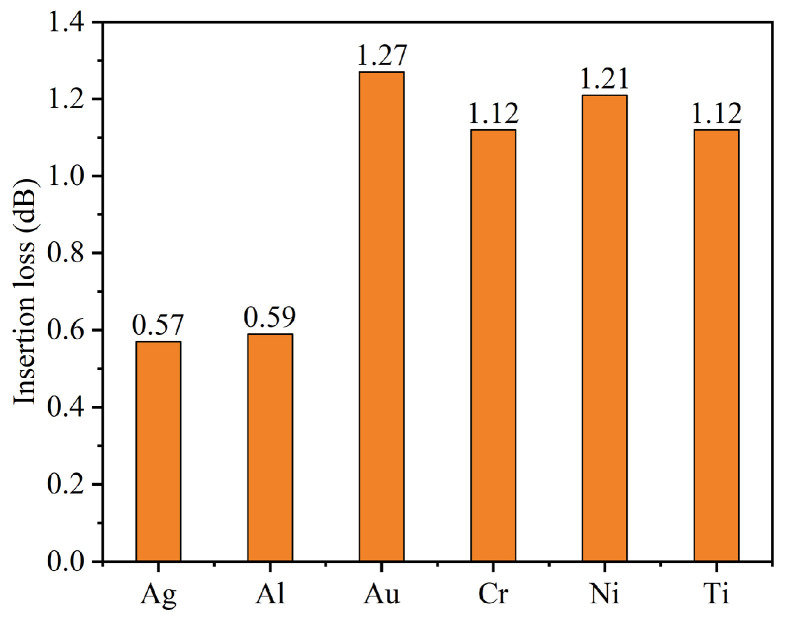
Insertion loss of the GaN bent waveguide with different metal films under the back-thinning optimization processes.

**Figure 9 nanomaterials-15-00151-f009:**
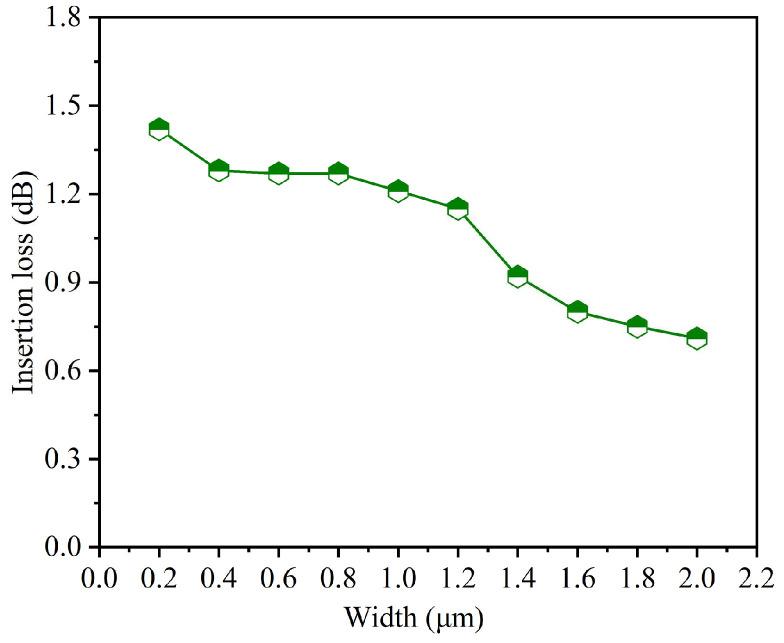
Insertion loss of the GaN bent waveguide with waveguide with the width ranging from 0.2 μm to 2.0 μm under the back-thinning optimization processes.

**Figure 10 nanomaterials-15-00151-f010:**
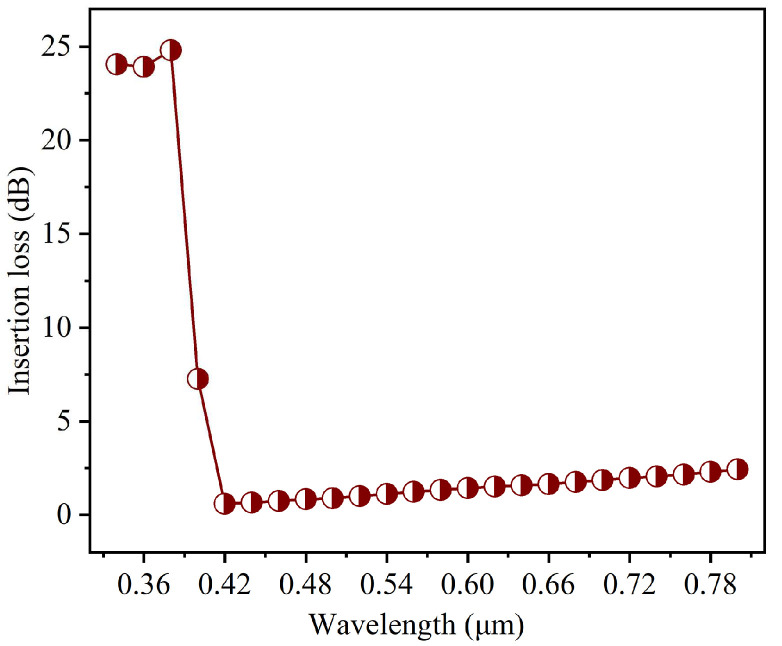
The wavelength dependence (ranging from 0.34 μm to 0.80 μm) of insertion loss for the back-side thinning optimized bent waveguide.

## Data Availability

Data are contained within the article, and further inquiries can be directed to the corresponding author.

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
