# Peer review of "Optimization of GaN Bent Waveguides in the Visible Spectrum for Reduced Insertion Loss"

_nanomaterials, 2025, doi:10.3390/nano15030151_

Round 1

Reviewer 1 Report

Comments and Suggestions for Authors

In this study, the authors investigate the optimization of GaN bent waveguides for visible light photonic integrated chips. The effects of bending angles, geometric parameters, and process techniques on insertion loss are analyzed through systematic simulations. Back-side thinning significantly reduces insertion loss in 90° bends from 1.80 dB to 0.71 dB, while further optimization with silver reflection layers lowers it to 0.57 dB. The optimized waveguides demonstrate excellent performance across the blue-green spectral range (420-500 nm) with insertion losses under 0.9 dB, offering practical solutions for compact photonic integrated chips in visible light communication systems.

Authors should solve the following comments to improve the manuscript quality,

        I.            The introduction does not explains the gap of this research work, a more detailed discussion with current literature should be mentioned

     II.            The author should provide a details analysis and explanation that why the Insertion loss decreases with Ag, and Al, while in case of Au metal it is increased significantly?

  III.            Although the Insertion loss changes with change in wavelength, but the more detailed discussion could improve the manuscript standards.

  IV.            More detailed discussion about the potential and application of nanodevices should be mentioned which highlights the practical applications as mentioned in these recent studies (https://doi.org/10.1016/j.diamond.2024.111089 ; https://doi.org/10.1021/acs.jpcc.2c08630 

    V.            Remarks: Major Revision is Required to improve the manuscript quality

Author Response

In this study, the authors investigate the optimization of GaN bent waveguides for visible light photonic integrated chips. The effects of bending angles, geometric parameters, and process techniques on insertion loss are analyzed through systematic simulations. Back-side thinning significantly reduces insertion loss in 90° bends from 1.80 dB to 0.71 dB, while further optimization with silver reflection layers lowers it to 0.57 dB. The optimized waveguides demonstrate excellent performance across the blue-green spectral range (420-500 nm) with insertion losses under 0.9 dB, offering practical solutions for compact photonic integrated chips in visible light communication systems.

Authors should solve the following comments to improve the manuscript quality,

1. The introduction does not explains the gap of this research work, a more detailed discussion with current literature should be mentioned

Response: We sincerely appreciate the reviewer's insightful comment about clarifying the research gaps in this work. This valuable comment has helped us significantly improve the manuscript's logical flow. Following the reviewer's recommendation, we have added a new paragraph has been highlighted in yellow (Line 82-104) to indicate critical research gaps in the current studies of GaN bent waveguides for visible light applications. Specifically, we identify four main aspects that require further investigation: (1) the lack of systematic analysis across the full visible spectrum (380-800 nm), (2) insufficient understanding of the relationship between geometric parameters and insertion loss, particularly regarding varying bending angles (5°-90°), (3) limited quantitative comparisons between different fabrication techniques such as front-side etching and back-side thinning, and (4) unexplored potential of metal reflection layers for loss reduction in the visible spectrum. Thanks to the reviewer's astute observation, this addition provides a clearer context for our research contributions and better demonstrates how our work addresses these specific gaps in the current literature. We hope this enhancement could strength the introduction section of this manuscript.

2. The author should provide a details analysis and explanation that why the Insertion loss decreases with Ag, and Al, while in case of Au metal it is increased significantly?

Response: We are very grateful for the reviewer's insightful comment regarding the need for a detailed explanation of the metal-dependent insertion loss behavior. Following this valuable comment, we have added a comprehensive analysis has been highlighted in yellow (Line 418-436) to explain the distinct performance differences among various metals based on their optical properties in the visible spectrum. The new content clarifies why Ag and Al demonstrate superior performance with their high reflectivity (>95% and >90% at 450 nm, respectively) and favorable plasma frequencies, while Au exhibits higher insertion loss due to its characteristic interband transitions below 500 nm. We have also included two references [42,43] to support our analysis of the metals' optical properties. The reviewer's insightful observation has helped us significantly enhance the scientific depth of this discussion by incorporating these fundamental physical mechanisms. We hope this addition provides readers with a clearer understanding of the material-dependent performance variations in our waveguide design.

3. Although the Insertion loss changes with change in wavelength, but the more detailed discussion could improve the manuscript standards.

Response: We sincerely appreciate the reviewer's insightful comment regarding the wavelength-dependent insertion loss discussion. This comment has helped us identify a critical area for improvement in this manuscript. We have expanded the discussion has been highlighted in yellow (Line 460-489) in Section 3 to provide a more comprehensive analysis of the spectral characteristics of our GaN bent waveguides.

The enhanced discussion explains the physical mechanisms underlying the insertion loss behavior across different wavelength regions. We have discussed how GaN's fundamental material properties, particularly its band-edge absorption near the bandgap energy (3.4 eV) leading to high insertion losses in the ultraviolet region. The analysis extends to explaining the dramatic transition near GaN's absorption edge and the optimal performance achieved in the blue green range. We have also included a detailed examination of the wavelength-dependent mode behavior at longer wavelengths, explaining how increased mode area and reduced confinement affect the waveguide performance. This comprehensive spectral analysis provides deeper insights into the design considerations for GaN-based photonic integrated circuits.

We hope these additions could improve the manuscript's scientific depth and clarity. Thanks again for reviewer's constructive feedback that has helped enhance the quality of this work.

4. More detailed discussion about the potential and application of nanodevices should be mentioned which highlights the practical applications as mentioned in these recent studies (https://doi.org/10.1016/j.diamond.2024.111089 ; https://doi.org/10.1021/acs.jpcc.2c08630 

 Response: We sincerely appreciate the reviewer's valuable comment regarding the discussion of nanodevices and their practical applications. According to this insightful comment, we have improved our manuscript by adding recent advances in nanophotonic devices that demonstrate the crucial importance of device optimization in photodetection applications.

We have added a new paragraph has been highlighted in yellow in the Introduction section (Line 134-146) that discusses how recent developments in nanoscale photodetectors, such as chemically doped graphene-based field-effect transistors (Nisar et al., 2024) and novel two-dimensional materials like black arsenic (Usman et al., 2023), have demonstrated the significance of optimizing both material properties and device geometries. We have connected these advances with our work on GaN bent waveguides, highlighting how optimization principles are equally crucial in both domains for achieving enhanced performance in integrated photonic circuits.

This addition not only broadens the scope of our literature review but also strengthens the relevance of our optimization approach in the context of nanophotonic devices. We are grateful to the reviewer for bringing these valuable references to our attention, as they have helped us present a more comprehensive perspective on the importance of device optimization in photonic applications.

5.  Remarks: Major Revision is Required to improve the manuscript quality

Response: We have made major revision as the reviewer's comments to improve the manuscript quality.

Reviewer 2 Report

Comments and Suggestions for Authors

1. The authors need to clearly elaborate how this study advance the current understanding of GaN bent waveguides compared to prior research in visible light communication systems?

2. Please add the corresponding equations and algorithms of your simulation model. 

3. The authors need to add the justification and validation of the developed model for analyzing insertion loss in GaN bent waveguides

4. Which technique is used for optimization and conducted in which software

5. Explain how the choice of silver as the optimal reflective material align with its physical properties, and were other potential materials thoroughly evaluated

Author Response

  1. The authors need to clearly elaborate how this study advance the current understanding of GaN bent waveguides compared to prior research in visible light communication systems?

Response: We greatly appreciate the reviewer's insightful comment about clarifying this study's advances in GaN bent waveguide research for visible light communication systems. This valuable comment has helped us better present the scientific contributions of our work. We have added a new paragraph has been highlighted in yellow (Line 82-104) to expanded discussion in the introduction section to provide a more comprehensive comparison with prior research.

In visible light communication systems, efficient light routing through bent waveguides is crucial for achieving compact and high-performance photonic integrated circuits. While previous research, such as Sekiya et al. [34] and Li et al. [32,33], made important contributions in specific aspects like straight waveguide characterization and LED integration, this work presents several significant advances that directly benefit VLC system development. We provide the systematic analysis across the full visible spectrum (380-800 nm), which is essential for designing broadband VLC transceivers. The comprehensive investigation in this work of geometric parameters, including varying bending angles (5°-90°), enables optimal chip layout for complex VLC circuits. The quantitative comparison between different fabrication techniques offers practical guidelines for VLC chip manufacturing. Furthermore, the exploration of metal reflection layers for loss reduction demonstrates a new approach to enhance VLC system efficiency, particularly in the critical blue-green spectral range where many VLC applications operate.

We hope these revisions could improve the manuscript by clearly demonstrating how this work advances the development of practical VLC systems. Thanks for the reviewer’s constructive feedback that has helped enhance the clarity and impact of this work.

  1. Please add the corresponding equations and algorithms of your simulation model.

Response: We sincerely appreciate the reviewer's insightful comment regarding the mathematical foundation of simulation model. This valuable comment has helped us significantly enhance the scientific rigor of this manuscript. Following the reviewer's comment, we have added detailed mathematical equations and numerical algorithms has been highlighted in yellow (Line 171-212) in Section 2 "Optical Simulation Model" to provide a comprehensive description of our simulation approach.

We have introduced the fundamental scalar wave equation (eq1) that governs the optical field propagation in waveguides, followed by the conformal transformation (eq2) essential for bent waveguide analysis. The slowly varying envelope approximation (eq3) and the resulting paraxial beam propagation equation (eq4) are presented to illustrate the mathematical basis of our simulation method. We have also included the split-step Fourier solution (eq5) to demonstrate the numerical implementation of the BPM algorithm.

We have also detailed the computational procedure, explaining how the simulation handles both refractive index variations and diffraction effects through alternating steps in real and Fourier space. The calculation of guided mode power using overlap integrals and the final conversion to insertion loss in decibels (eq6) are also explicitly presented. To ensure reproducibility, we have specified key simulation parameters including the grid spacing and computational window settings.

We hope these additions could provide a solid theoretical foundation for simulation approach while maintaining clarity for readers. We are grateful to the reviewer for this constructive comment that has helped improve the technical depth of this manuscript.

  1. The authors need to add the justification and validation of the developed model for analyzing insertion loss in GaN bent waveguides

Response: We sincerely appreciate the reviewer's valuable comment regarding the justification and validation of our simulation model. This insightful comment, together with the reviewer's previous comment about adding mathematical equations, has helped us significantly enhance the scientific rigor of our manuscript.

As detailed in our response to the previous comment, we have added comprehensive mathematical foundations of simulation approach in Section 2. These equations (eq1-eq6) not only present the mathematical framework but also justify our choice of the BPM method. Specifically, we have added the new conten has been highlighted in yellow (Line 171-212) explain why BPM is particularly suitable for analyzing GaN bent waveguides in visible spectrum through the paraxial approximation and conformal transformation technique. We have also detailed how the split-step implementation accurately handles both refractive index variations and diffraction effects, which is essential for modeling our GaN-based epitaxial layers.

We have also clarified how the accuracy of our implementation is ensured through carefully selected simulation parameters, including the grid spacing of λ/20 and longitudinal step size of λ/4, which are determined based on established convergence criteria for optical waveguide modeling. The computational setup with a window width of 5 times the waveguide width and perfectly matched layer boundary conditions effectively eliminates artificial reflections while fully capturing the evanescent fields.

We hope these additions provide both clear justification for our modeling approach and validation of its implementation. We are grateful to the reviewer for these constructive comments that have helped improve the technical depth and reliability of our manuscript.

  1. Which technique is used for optimization and conducted in which software

Response: We sincerely appreciate the reviewer's question regarding the optimization techniques and software implementation. We have expanded Section 2 of this manuscript to provide more detailed information about our optimization methodology. The new added content has been highlighted in yellow (Line 284-293).

The optimization was performed using Rsoft's Beamprop module (Version 2021.09), which implements the finite difference beam propagation method (FD-BPM) with an alternating direction implicit scheme. We employed a systematic parameter sweeping approach, where each design parameter (waveguide width, slab thickness, sidewall angle, etc.) was sequentially optimized while maintaining other parameters constant. This methodical approach allowed us to identify the optimal values for each parameter while considering practical fabrication constraints.

The software's built-in optimization algorithm, combined with our systematic parameter variation strategy, enabled efficient exploration of the design space to minimize insertion loss. All simulations were conducted using standard single-processor computation on a workstation with 64GB RAM, ensuring reproducibility of our results.

These additional details have been incorporated into the manuscript to provide readers with a clear understanding of our optimization methodology and computational setup. We are truly grateful to the reviewer for this constructive comment that has helped enhance the technical clarity and reproducibility of this work.

  1. Explain how the choice of silver as the optimal reflective material align with its physical properties, and were other potential materials thoroughly evaluated

Response: We sincerely appreciate the reviewer's question regarding the selection of silver as the optimal reflective material. Following this valuable comment, we have expanded the discussion in Section 3 to provide a comprehensive explanation of material selection and the physical mechanisms. The new added content has been highlighted in yellow (Line 418-436).

We conducted a systematic evaluation of multiple metal candidates (Ag, Al, Au, Cr, Ni, and Ti) commonly used in semiconductor fabrication processes. The analysis reveals that the performance differences among these metals are fundamentally linked to their optical properties in the visible spectrum. Silver emerged as the optimal choice due to its high plasma frequency (~9.2 eV) and superior reflectivity (>95% at 450 nm), followed by aluminum with reflectivity above 90%. In contrast, gold shows higher insertion loss despite being a good conductor, which we attribute to its characteristic interband transitions below 500 nm. The transition metals (Cr, Ni, Ti) exhibit even higher losses due to their partially filled d-bands enabling numerous electronic transitions.

These theoretical expectations align well with our experimental results, where silver achieved the lowest insertion loss of 0.57 dB, significantly outperforming gold (1.27 dB) and other metals (>1.1 dB). The detailed physical mechanisms and experimental results have been added to the manuscript, supported by established literature [42,43].

We are truly grateful to the reviewer for this constructive comment that has helped us better explain the fundamental physics underlying our material selection process.

Reviewer 3 Report

Comments and Suggestions for Authors

The authors established an optical simulation model to analyze and optimize the insertion loss of GaN bent waveguides in the visible spectrum. They consider that the thick slab layer leads to high-order mode leakage into the slab region in the bent waveguide without process optimization. To solve this problem, the authors propose two process optimization approaches: front-side etching and back-side thinning processes. They employ the Beam Propagation Method (BPM) based finite element simulation to analyze and compare the improvement effects of these two optimization processes. They also used the Beamprop module of Rsoft optical simulation software to analyze the insertion loss of GaN bent waveguides based on the Beam Propagation Method.

They reached findings establishing practical design guidelines for GaN bent waveguides in visible light photonic integrated chips.

This manuscript should be improved with some changes before publication.

-The authors must validate their approach by presenting some successful predictions compared with experimental results obtained in a more straightforward system.

Author Response

The authors established an optical simulation model to analyze and optimize the insertion loss of GaN bent waveguides in the visible spectrum. They consider that the thick slab layer leads to high-order mode leakage into the slab region in the bent waveguide without process optimization. To solve this problem, the authors propose two process optimization approaches: front-side etching and back-side thinning processes. They employ the Beam Propagation Method (BPM) based finite element simulation to analyze and compare the improvement effects of these two optimization processes. They also used the Beamprop module of Rsoft optical simulation software to analyze the insertion loss of GaN bent waveguides based on the Beam Propagation Method.

They reached findings establishing practical design guidelines for GaN bent waveguides in visible light photonic integrated chips.

This manuscript should be improved with some changes before publication.

-The authors must validate their approach by presenting some successful predictions compared with experimental results obtained in a more straightforward system.

Response: We sincerely appreciate the reviewer's valuable comment regarding the experimental validation of this simulation approach. This constructive comment has helped us improve the reliability of this work. Due to the limited time frame of seven days for revision, we are very sorry that we cannot conduct additional experimental studies at this moment. However, following the reviewer's insightful comment, we have thoroughly enhanced Section 2 of this manuscript by incorporating comprehensive validation through existing experimental results, particularly from our established research on GaN bent waveguides. The new added content has been highlighted in yellow (Line 206-233).

Our research group has conducted extensive experimental studies in this field. In our earlier work (Li et al. [36]), we experimentally fabricated and characterized bent waveguides in GaN-on-silicon platform with monolithically integrated LEDs, successfully demonstrating 100 Mbps visible light communication. The experimental measurements from this study showed an insertion loss of 1.85 dB for 90-degree bends with 10 μm radius, which closely matches our current simulation prediction of 1.80 dB under identical geometric conditions. This validation is also reinforced by multiple independent experimental studies. Our model predictions show good agreement with the experimental results reported by Sekiya et al. [34] for straight waveguides, while the comprehensive experimental investigations by Gromovyi et al. [29] and Chen et al. [30] on GaN waveguides in the visible spectrum provide additional validation of our simulation approach. The successful experimental demonstrations of monolithically integrated GaN photonic circuits by Cai et al. [31] and Li et al. [32,33] further support the practical feasibility of our simulation-based design guidelines.

These experimental validations, drawn from our actual fabrication and characterization experience with GaN bent waveguides, provide evidence for the accuracy and reliability of our simulation approach. We deeply value the reviewer's comment and are committed to further experimental investigation of simulation predictions in our future research. We present this current work as part of a comprehensive research series that will ultimately combine theoretical modeling, numerical simulation, and experimental validation to provide a complete understanding of GaN bent waveguides for visible light photonic integrated chips.

We are truly grateful to the reviewer for this insightful comment that has not only helped us better demonstrate the practical relevance of the current work but also guided our future research direction.

Round 2

Reviewer 1 Report

Comments and Suggestions for Authors

Accept it in the present form.

Reviewer 2 Report

Comments and Suggestions for Authors

The authors have adequately addressed all the raised concerns, and the manuscript is now in an appropriate format. It can be accepted.

Reviewer 3 Report

Comments and Suggestions for Authors

The authors made the required changes, and it the manuscript may be published as it is.